**communications**

biology

# Multiplexed labeling of cellular proteins with split fluorescent protein tags

Ryo Tamura[1], Fangchao Jiang[2], Jin Xie [2] & Daichi Kamiyama [1✉]

Self-complementing split fluorescent proteins (split $FP_{1-10/11}$) have become an important labeling tool in live-cell protein imaging. However, current split FP systems to label multiple proteins in single cells have a fundamental limitation in the number of proteins that can be simultaneously labeled. Here, we describe an approach to expand the number of orthogonal split FP systems with spectrally distinct colors. By combining rational design and cycles of directed evolution, we expand the spectral color palette of $FP_{1-10/11}$. We also circularly permutate GFP and synthesize the β-strand 7, 8, or 10 system. These split GFP pairs are not only capable of labeling proteins but are also orthogonal to the current $FP_{1-10/11}$ pairs, offering multiplexed labeling of cellular proteins. Our multiplexing approach, using the new orthogonal split FP systems, demonstrates simultaneous imaging of four distinct proteins in single cells; the resulting images reveal nuclear localization of focal adhesion protein Zyxin.

[1] Department of Cellular Biology, University of Georgia, Athens, GA, USA. [2] Department of Chemistry, University of Georgia, Athens, GA, USA. ✉email: daichi. kamiyama@uga.edu

In the self-complementing split GFP system, super-folder GFP is split between β-strands 10 and 11, rendering 214-amino acid and 16-amino acid fragments[1]. The short fragment, $GFP_{11M3}$ $_{OPT}$, acts as an epitope tag when inserted into a gene of interest[2]. When expressed in the same cell, the $GFP_{1-10\ D7}$ and $GFP_{11M3}$ $_{OPT}$ fragments (hereafter referred to as $GFP_{1-10/11}$) spontaneously interact with each other to form a functional GFP (Supplementary Fig. 1). The $GFP_{11}$ fragment has been used in numerous biological studies[3,4]: targeting nanomaterials in cells[5,6], forming protein oligomeric structures[2,7], verifying aggregation processes[8], and imaging protein localization in living cells[9]. By introducing additional substitutions into the $GFP_{1-10}$ fragment, cyan and yellow spectral variants were previously created and used to visualize localization patterns of cellular proteins[2,10]. The majority of substitutions which lead to the spectral shifts in these variants are located within the large fragments (i.e., $CFP_{1-10}$ and $YFP_{1-10}$). These fragments retain the ability to bind to the identical $GFP_{11}$ fragment, so that reconstitution with $GFP_{11}$ produces a functional cyan or yellow FP.

These self-complementing split GFP variants have already become a powerful and versatile tool for various imaging applications. In particular, endogenously tagged cell lines can be produced by the efficient introduction of the short fragment ($GFP_{11}$) into a genomic locus without perturbing local genomic structure[2,11]. Additionally, we have been able to generate a library of human cells with $GFP_{11}$-tagged endogenous proteins via CRISPR/Cas9-mediated homology-directed repair (HDR), and demonstrate that $GFP_{11}$-tag is compatible with a wide range of cellular proteins such as enzymes, receptors, transport proteins, and structural proteins[12]. However, labeling multiple proteins simultaneously in single cells has been challenging. Multiplexed visualization is tremendously beneficial for simultaneous comparisons of protein dynamics. Recently, great advances have been made in split super-folder Cherry ($sfCherry_{1-10/11}$) as a second, orthogonal split FP system[2,13,14]. The $GFP_{11}$ and $sfCherry_{11}$ fragments allow simultaneous labeling of two different proteins. Although this multicolor approach has expanded the potential of split-FP labeling, it has a bottleneck in multiplexing because of the limited number of available orthogonal split FP systems with different colors.

In this report, we expand the color palette of self-associating split FPs. We have introduced rational mutations into the amino acid sequence of EBFP2 through site-directed mutagenesis and generated two blue-colored split FPs, $EBFP2_{1-10/11}$, and $Capri_{1-10/}$ $_{11}$. We have also engineered self-associating fragments of mRuby3 (mRuby3 is a red-colored FP with a shorter-wavelength than sfCherry)[15]. We have evolved $mRuby3_{1-10}$ by a directed evolution strategy to increase its complementation with $mRuby3_{11}$. Our final optimized construct, split mRuby4, becomes a fusion pair when expressed in human cells. In addition, we propose a new approach to generate more orthogonal split FPs using circularly permutated FP fragments. This approach can potentially overcome multiplexing limitations of split-FP labeling. Finally, as a proof-of-concept experiment, we applied our technique to visualize differential distribution of four proteins in single human cells and found that focal adhesion protein Zyxin sometimes accumulated in the nucleus.

## Results and discussion

**Rationally designed variants of split BFP and CFP.** To expand our color palette of split FPs, we split EBFP2 at the same site as $GFP_{1-10/11}$ (note that EBFP2 is 4-fold brighter and >500-fold more photostable than EBFP[16]). While the short fragment is identical to the amino acid sequence of $GFP_{11}$, six substitutions have been introduced into the large fragment through site-

directed mutagenesis (N40I/T106K/E112V/K166T/I167V/S206T; the numbering of amino acids follows that of EBFP2). These substitutions have been previously shown to enhance complementation efficiency of $GFP_{1-10}$ variants[1]. To verify in vivo complementation between the two fragments, we used $GFP_{11}$-tagged β-actin and histone 2B. Co-expressing each one with $EBFP2_{1-10}$ in HeLa cells, we observed blue fluorescence in images of the actin stress fibers and the nucleoplasm (Fig. 1a, b). In some cases (e.g., the actin image), autofluorescence limits the usefulness of this split construct because its overall fluorescent signal is extremely weak. In fact, high autofluorescence background with UV light is often observed in the perinuclear region (Supplementary Fig. 2). To improve its overall brightness, we decided to add six more substitutions to $EBFP2_{1-10}$ (S65T/Q80R/F99S/ V128T/M153T/V163A; some of these have previously been characterized to promote the stability and folding rate of GFP[1,17]). This new split FP, termed split Capri for its cyan–blue color, has the same absorption spectrum as split EBFP2 (Supplementary Fig. 3). The emission spectrum, however, is red-shifted from split EBFP2 by 20 nm ($\lambda_{abs}/\lambda_{em} = 379/469$ nm). Furthermore, its peak extinction coefficient of 37,300 $M^{-1}\,cm^{-1}$ and quantum yield of 0.13, are greater than those of split EBFP2 (Supplementary Table 1). When associated with $GFP_{11}$-tagged β-actin or H2B, $Capri_{1-10}$ exhibits very bright fluorescence in HeLa cells (Fig. 1c, d). To assess the improvement in the resulting brightness, we co-expressed $GFP_{11}$-H2B in HEK 293T cells with either $EBFP2_{1-10}$ or $Capri_{1-10}$. Quantifying the fluorescence intensity of cells by flow cytometry, we found that $Capri_{1-10/11}$ had a four-fold brighter fluorescence than $EBFP2_{1-10/11}$ (Supplementary Fig. 4).

In addition to BFP variants, cyan-colored FPs have been widely studied. When we introduced substitutions into $GFP_{1-10}$ (Y66W to make $CFP_{1-10}$), complementation fluorescence was observed for $GFP_{11}$-β-actin or $GFP_{11}$-H2B fusions co-expressing with $CFP_{1-10}$ in HeLa cells (Supplementary Fig. 5). However, it is

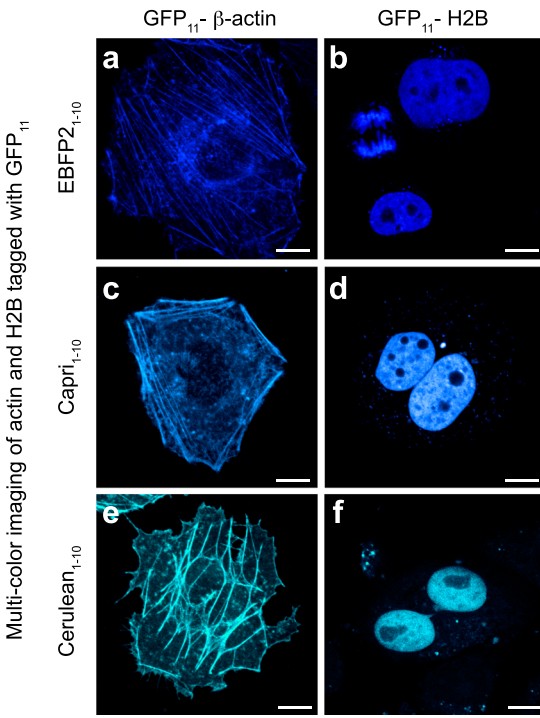

**Fig. 1 Performance of BFP and $CFP_{1-10/11}$ variants in fusion constructs. a–f** Confocal images of HeLa cells. Cells co-expressing $EBFP2_{1-10}$ (**a**, **b**), $Capri_{1-10}$ (**c**, **d**), and $Cerulean_{1-10}$ (**e**, **f**) with $GFP_{11}$-tagged β-actin or histone 2B.

noticeable in the figure that the overall brightness of $CFP_{1-10}$ is relatively weak, making it difficult to visualize thin actin filaments (A recent in vitro assessment also reported that split CFP has a low brightness[10]). Therefore, we sought to produce a cyan-colored FP that has enhanced brightness. A recent improvement of full-length CFP, named Cerulean, increases the brightness by ~1.6 times[18]. Because the known substitutions are located only on $GFP_{1-10}$ (Y66W/S72A/Y145A/H148D for Cerulean), $Cerulean_{1-10}$ can associate with $GFP_{11}$. To evaluate the enhancement in its overall brightness for cellular microscopy, we prepared plasmids encoding $Cerulean_{1-10}$ or $CFP_{1-10}$. We co-transfected each one of these plasmids with a $GFP_{11}$-H2B plasmid in HEK 293T cells. Imaging by confocal microscopy, we quantified the signal level of these split FPs. We found that $Cerulean_{1-10}$ signal was ~1.7 times brighter than that of $CFP_{1-10}$ ($p < 0.0001$, Student's $t$-test; Supplementary Fig. 6). We next assessed the performance of $Cerulean_{1-10}$ when used as a fusion tag. We used $GFP_{11}$-fused β-actin or H2B and co-expressed each one with $Cerulean_{1-10}$ in HeLa cells (Fig. 1e, f). Although we observed complementation of $Cerulean_{1-10}$ in the appropriate locations, some cells exhibited thicker actin bundles, which we have never seen in cells expressing a full-length Cerulean fusion (Fig. 1e and Supplementary Fig. 7). Because this artifact is common for dimeric or tetrameric FPs when they are targeted to two-dimensional structures[19,20], we suspect that $Cerulean_{1-10/11}$ is an oligomeric split FP. Nonetheless, an in-depth investigation is required to validate such a property in split Cerulean and under more various experimental conditions.

**Engineering of a red-colored split FP variant based on mRuby3.** Although developmental efforts are ongoing to improve the brightness of split sfCherry[2,13,14], having spectrally distinct split red FPs would foster the gross usefulness of $FP_{11}$-tags. Since split sfCherry2 has a far-red shifted emission peak at 610 nm, we sought to explore the evolution of orange-red FPs such as mKO2, mRuby3, mApple, and mScarlet-I[15,21–23] in E. coli. Following the previously established approach[13], we inserted a 30 amino-acid spacer between the 10th and 11th β-strand of the four FPs. The long spacer insertion greatly diminished colony fluorescence of mKO2, mApple, and mScarlet-I, while colonies expressing spacer-inserted mRuby3 remained fluorescent (Supplementary Fig. 8). To improve the brightness of the spacer-inserted mRuby3, we mutagenized it using error-prone PCR and then transformed into E. coli; the three brightest candidates were pooled and subjected to another round. After three rounds, brightness of the best candidate revamped six-fold relative to that of spacer-inserted mRuby3 (Supplementary Fig. 9). We found seven substitutions in $mRuby3_{1-10}$ (M15T/Q27H/T31I/V106I/S113C/R126S/A154V) and termed this variant split mRuby4 ($\lambda_{abs}/\lambda_{em} = 557/592$ nm; see also Supplementary Fig. 3 for its absorbance spectrum). Compared to split mRuby3, we created a particularly bright variant that has a higher extinction coefficient and increased quantum yield (Supplementary Table 1).

To assess whether split mRuby4 could fluoresce in human cells, we over-expressed $mRuby4_{11}$-β-actin with $mRuby4_{1-10}$ in HeLa cells. We observed that complemented split mRuby4 has a bright signal in fluorescent images of actin and various fusion proteins (Fig. 2a–f). To determine the signal level of split mRuby4, we performed a cellular fluorescence measurement by flow cytometry and compared the signal to full-length mRuby3. With expression of spacer-inserted mRuby4 in HEK 239T cells, we found that its signal level became around 69% of full-length mRuby3 (Supplementary Fig. 10). We have also demonstrated that $mRuby4_{1-10/11}$ has sufficient efficiency to detect proteins expressed at endogenous levels. We employed CRISPR/Cas9-mediated HDR and

introduced a 200-nucleotide ssDNA donor into the *HIST2H2BE* locus of HEK 293FT cells expressing $mRuby4_{1-10}$. Subsequently, we found that split mRuby4 complementation had a prominent signal in images of the $mRuby4_{11}$ knock-in (Supplementary Fig. 11).

As shown in Fig. 2g, the emission peaks for split mRuby4 and split sfCherry2 are only 20 nm apart yet still visually distinguishable (Supplementary Fig. 12). To further evaluate how many split FPs could be simultaneously visualized in different cells, we performed spectral imaging of HEK 293 cells expressing H2B fused proteins (Fig. 2g). We co-cultured six types of HEK 293 cells, each of which expressed H2B labeled with either $EBFP2_{1-10/11}$, $Capri_{1-10/11}$, $Cerulean_{1-10/11}$, $GFP_{1-10/11}$, $mRuby4_{1-10/11}$, or $sfCherry2_{1-10/11}$. After we synchronized at the G2/M phase by release from a cyclin-dependent kinase inhibitor, we imaged the cells (Supplementary Fig. 13). Within a couple of hours, 20% of the cell population was in cytokinesis, which is consistent with previous literature[24]. We then captured cells with each split FP fusion at different stages of mitosis (Fig. 2h). Overall, these experiments illustrate six-color spectral imaging of cellular proteins.

**Evaluation of split $FP_{1-10/11}$ systems for multiplexed imaging in single cells.** An orthogonal interaction between $GFP_{1-10/11}$ and $sfCherry2_{1-10/11}$—meaning $GFP_{11}$ can interact with $GFP_{1-10}$, but cannot interact with $sfCherry2_{1-10}$—is the basis of simultaneous labeling of two different fusion proteins in single cells. With an array of $FP_{1-10/11}$ pairs developed, we sought to systematically test their binding specificities by flow cytometry. $GFP_{1-10/11}$, $sfCherry2_{1-10/11}$, $mNeonGreen2_{1-10/11}$[13], and $mRuby4_{1-10/11}$ were examined for complementation in HEK 293T cells. Each $FP_{1-10}$ fragment was co-expressed with any one of four $FP_{11}$-fused β-actin, and the interactions were tested along the grid diagonal (Fig. 3a). As shown in Fig. 3a, all four $FP_{1-10}$ fragments reconstituted with their corresponding partners. Interestingly, $mRuby4_{1-10}$ and $sfCherry2_{11}$ formed complementation signal as did $mRuby4_{11}$ and $sfCherry2_{1-10}$. Because the $FP_{1-10/11}$ fragments encoded by closely related FPs, we expected there to be some crosstalk (Fig. 3b). We used HeLa cells co-expressing $GFP_{1-10/11}$-β-actin with Zyxin-$mRuby4_{1-10/11}$, or $mNeonGreen2_{1-10/11}$-β-actin with $mRuby4_{1-10/11}$-Clathrin to verify dual-color labeling with $mRuby4_{11}$ in single cells. We found that the two distinct fluorescence channels did not overlap in the cells (Fig. 3c, d).

**A strategy to create new orthogonal split FPs using circularly permuted FP fragments.** In order to provide more variants of split FPs orthogonal to existing $FP_{1-10/11}$, we took advantage of circularly permuted GFP variants[25]. By linking the N-termini and C-termini and cutting out a single β-strand, any one of the eleven β-strands could be a split GFP-tag. We chose to measure complemented signal of the β-strands 7, 8, 9, and 10. (The β-strands 1–6 were excluded because the complementary fragments of these strands are unlikely to be water-soluble[26]). To this end, we prepared DNA constructs encoding each of the β-strands fused to β-actin and measured the overall complemented signal of each construct in HEK 293T cells by flow cytometry. We observed fluorescence signal reconstituted from the β-strands 7, 8, and 10 (hereafter named $GFP_{8-6/7}$, $GFP_{9-7/8}$, and $GFP_{11-9/10}$) with their corresponding partners (Fig. 4a). These split GFPs retained 7–57% brightness of $GFP_{1-10/11}$, albeit leaving room for improvement. To validate protein labeling using the β-strands, we generated constructs encoding various cellular proteins fused with $GFP_8$ and co-expressed each one of them with $GFP_{9-7}$ in HeLa cells. For three proteins tested, we observed their expected localizations (Fig. 4b–d).

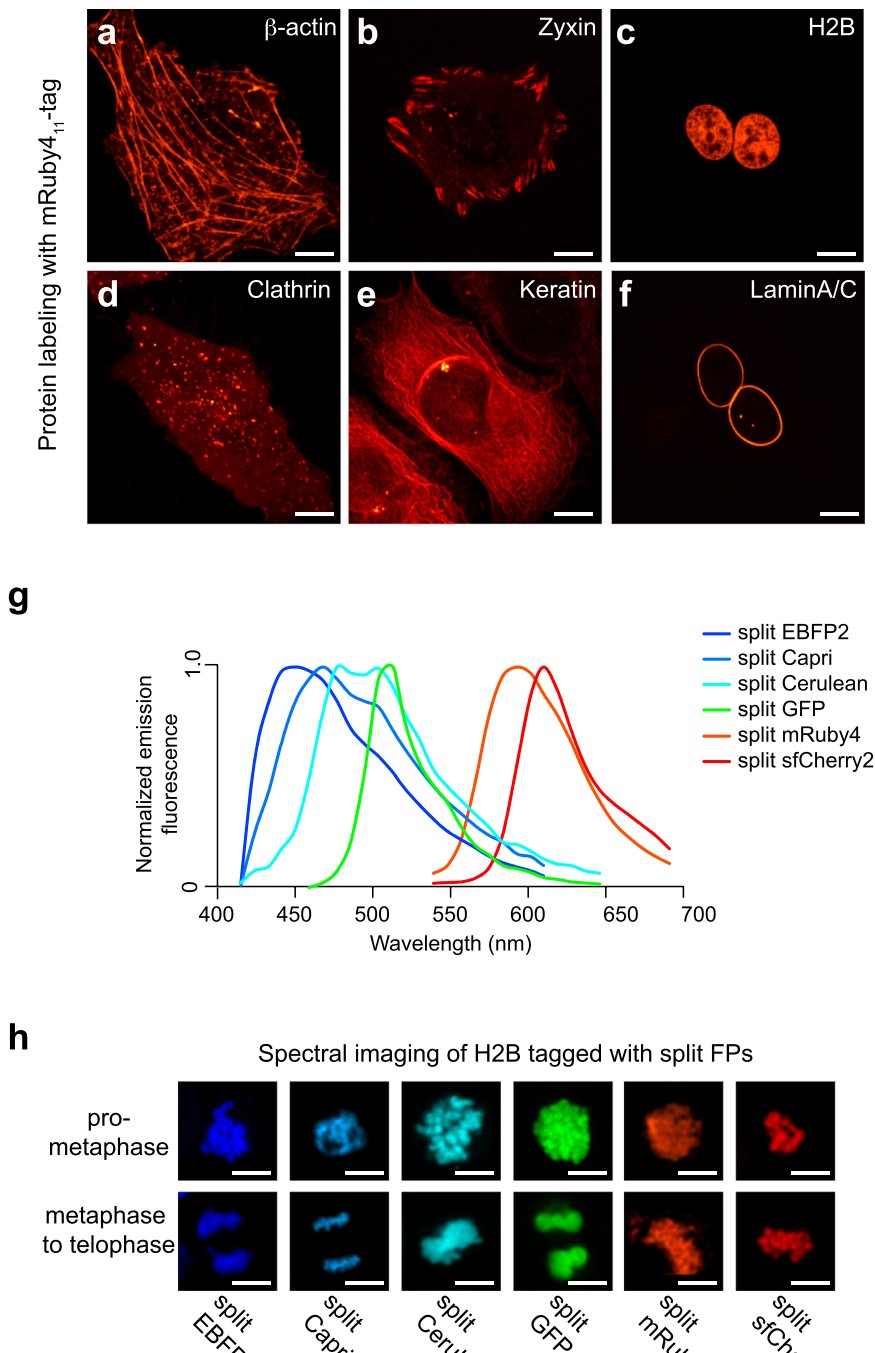

**Fig. 2 Development of mRuby4, a new red-colored split FP. a–f** Cells co-expressing mRuby4$_{1-10}$ with mRuby4$_{11}$-fused cellular proteins. For each, the name of the fusion partner and its normal subcellular location are indicated, respectively; β-actin, actin stress fibers (**a**); Zyxin, focal adhesion (**b**); histone 2B, nuclei (**c**); Clathrin light chain, clathrin-coated pits (**d**); Keratin, intermediate filaments (**e**); Lamin A/C, nuclear envelops (**f**). **g** Normalized fluorescence emission spectra of FP$_{1-10/11}$ variants in HeLa cells. (**h**) HEK 293 cells expressing H2B labeled with EBFP2$_{1-10/11}$, Capri$_{1-10/11}$, Cerulean$_{1-10/11}$, GFP$_{1-10/11}$, mRuby4$_{1-10/11}$, or sfCherry2$_{1-10/11}$ were co-cultured in the same plate. Spectrally unmixed images at the different stages of mitosis are represented (see also Supplementary Fig. 14). Scale bars,10 μm.

Next, we assessed the binding specificities of GFP$_{8-6/7}$, GFP$_{9-7/8}$, GFP$_{11-9/10}$, and GFP$_{1-10/11}$. We performed the flow cytometry assay conducted in a grid format as described earlier. Either GFP$_{8-6}$, GFP$_{9-7}$, GFP$_{11-9}$, or GFP$_{1-10}$ was co-expressed with β-actin fused with the β-strands 7, 8, 9, or 11 in HEK 293T cells. In this experiment, each of the β-strands only binds to its corresponding partner (Fig. 4e). For instance, GFP$_8$ interacts with GFP$_{9-7}$, but not

GFP$_{1-10}$. This orthogonal interaction was validated by dual-color imaging of U2OS cells, in which GFP$_{11}$-H2B and GFP$_8$-Lamin A/C were co-expressed with Capri$_{1-10}$ and GFP$_{9-7}$. We observed the exclusion of GFP$_8$-Lamin A/C from the nucleoplasm where GFP$_{11}$-H2B predominately localized (Fig. 4f). Taken together, circularly permuted FP fragments can be used to generate additional orthogonal pairs for multiplexed split FP-labeling.

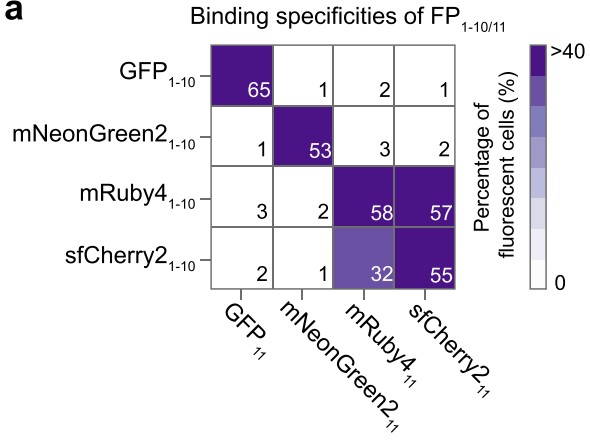

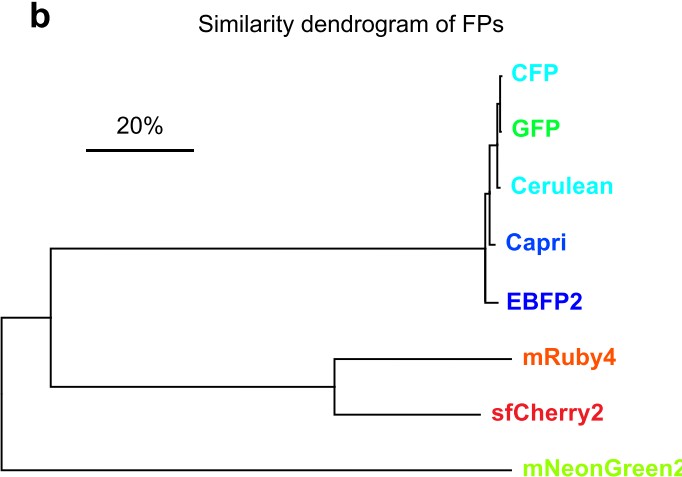

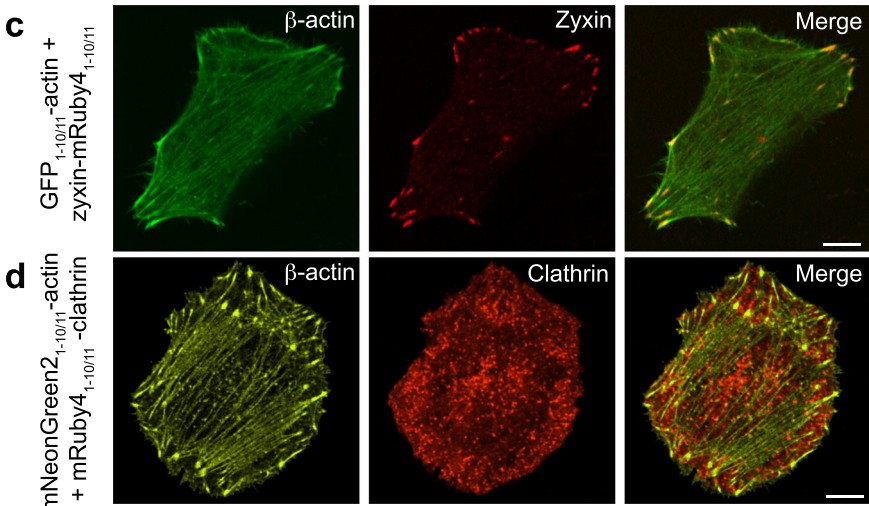

**Fig. 3 Characterizing the binding specificities of available FP$_{1\text{-}10/11}$ pairs. a** Characterizing the binding specificities of GFP$_{1\text{-}10/11}$, sfCherry2$_{1\text{-}10/11}$, mNeonGreen2$_{1\text{-}10/11}$, and mRuby4$_{1\text{-}10/11}$ by flow cytometry (see also Supplementary Fig. 15). Each of the FP$_{11}$ fragments was tested for complementation to all of the FP$_{1\text{-}10}$ fragments. Complementation is indicated as the percentage of fluorescent cells by a color scale and the number in each block. **b, c** Dual-color fluorescence images of HeLa cells expressing GFP$_{1\text{-}10/11}$-β-actin and Zyxin-mRuby4$_{1\text{-}10/11}$ (**b**), and mNeonGreen2$_{1\text{-}10/11}$-β-actin and mRuby4$_{1\text{-}10/11}$-Clathrin (**c**). **d** This dendrogram is based on the similarities of the following fluorescence protein sequences: EBFP2, Capri, Cerulean, CFP, GFP, mNeonGreen2, mRuby4, and sfCherry2. Proteins that share sequences are separated by smaller branch lengths. Scale bar, 20% dissimilarity. The dendrogram was constructed using MEGA 7.0 software. Scale bars, 10 μm.

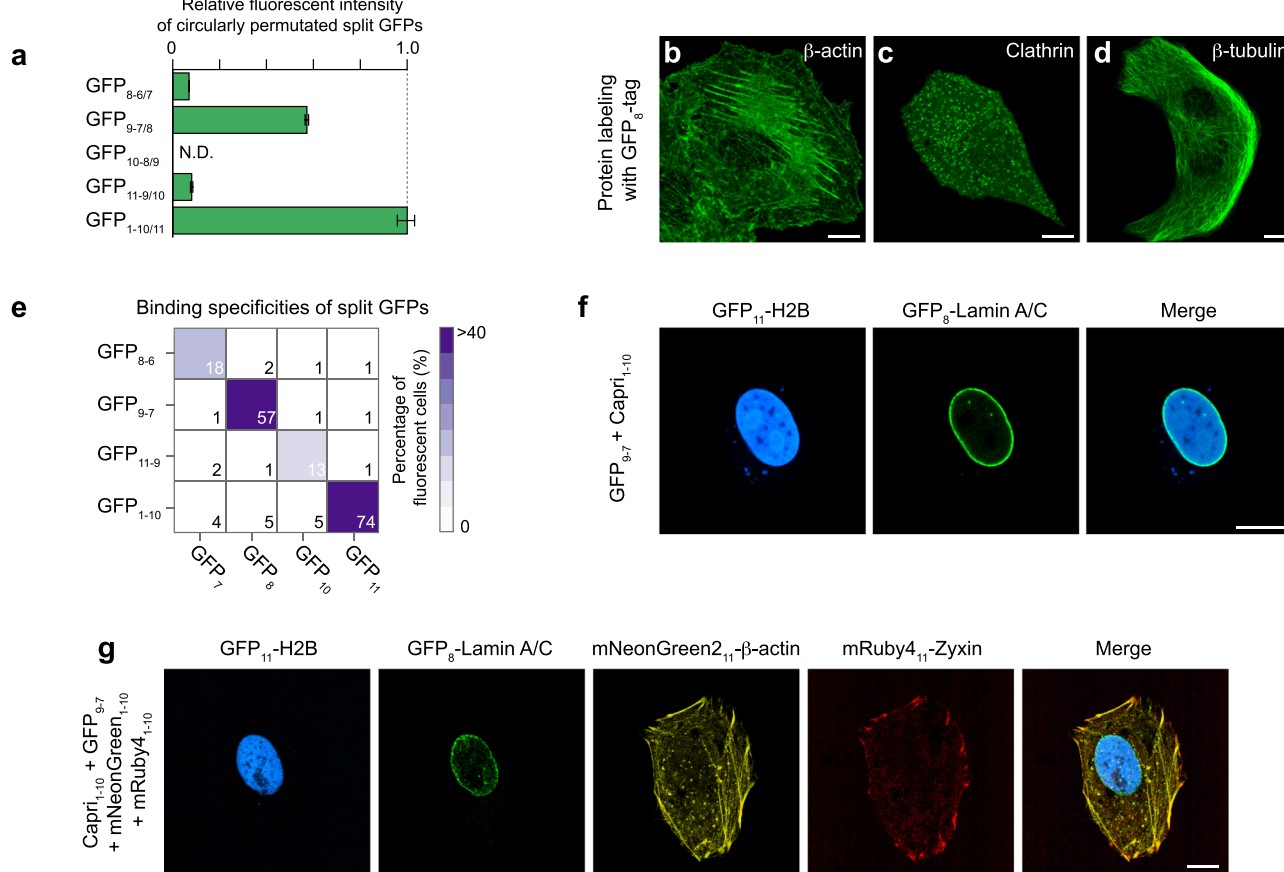

**Fig. 4 Multiplexed labeling of cellular proteins in human cells. a** Fluorescence intensity of HEK 293T cells expressing actin labeled with circularly permutated split GFP variants, measured by flow cytometry (see also Supplementary Fig. 16). $n = 698$ cells for $GFP_{8-6/7}$; $n = 7792$ for $GFP_{9-7/8}$; $n = 274$ for $GFP_{11-9/10}$; $n = 11017$ for $GFP_{1-10/11}$. Error bars are SEM. **b–d** Confocal images of HeLa cells co-expressing $GFP_{9-7}$ with $GFP_8$ fusions; β-actin (**b**), Clathrin light chain (**c**), and β-tubulin (**d**). **e** The binding specificities of $GFP_{8-6/7}$, $GFP_{9-7/8}$, $GFP_{11-9/10}$, and $GFP_{1-10/11}$ were characterized by flow cytometry (see also Supplementary Fig. 17). **f** Dual-color fluorescence images of a U2OS cell expressing two different fusions, $GFP_{11}$-H2B and $GFP_8$-Lamin A/C. **g** Four-color images of a U2OS cell co-expressing $GFP_{11}$-H2B, $GFP_8$-LaminA/C, mNeonGreen2$_{11}$-β-actin, and mRuby4$_{11}$-Zyxin with $GFP_{1-10}$, $GFP_{9-7}$, mNeonGreen2$_{1-10}$, and mRuby4$_{1-10}$. Scale bars, 10 μm.

**Multicolor images reveal nuclear localization of Zyxin**. Finally, we assessed the potential of split FP systems for multiplexed labeling of proteins in single cells. As a proof-of-principle, we used four orthogonal split FPs that we thoroughly investigated in this report (Capri$_{1-10/11}$, $GFP_{9-7/8}$, mNeonGreen2$_{1-10/11}$, and mRuby4$_{1-10/11}$), which are distinguishable from each other by using spectral imaging methodology (Fig. 4g). U2OS cells were transfected to express these split FPs targeted to four distinct proteins (H2B, Lamin A/C, β-actin, and Zyxin), and we observed their correct localization. Strikingly, a few cells displayed some portion of Zyxin proteins localized to the nucleus, although the proteins predominantly localized at focal adhesions in these cells. By visual inspection of a total of 145 cells, we found that 37 of these cells exhibited nuclear localization of Zyxin during interphase (Supplementary Fig. 18). Because Zyxin is a relatively large molecule (>80 kDa) but does not have a nuclear localization signal, Zyxin must enter the nucleus in contact with other components. We observed a similar nuclear localization pattern of Zyxin tagged with a full-length FP tag in U2OS cells (50 out of 174 cells), and found that this observation has also been confirmed in other cell lines[27–29].

For the initial demonstration of multiplexed labeling, split FPs were over-expressed as fusions to target proteins in single cells (Fig. 4g). However, these fusion proteins might be subject to limitations because of the potential for overexpression artifacts (e.g., aberrant organelle and/or cellular morphology). To further verify our observation in the future, this approach will be extended to label endogenous proteins by methods such as CRISPR/Cas9-mediated gene knock-in[12]. Because a split FP tag is ~42–63 nucleotides long (which is ~10 times smaller than the size of an intact FP), a short donor oligo can be directly synthesized, making this a cloning free approach (see also Supplementary Fig. 11)[2,12]. In addition, a small tag such as $GFP_{11}$-tag can be introduced into a host cell genome at high homologous recombination efficiencies[11]. Such a simple and efficient approach would facilitate the generation of multiple insertions in single cells. Thus, the sequences for multiple split FP tags could then be inserted either sequentially or simultaneously into targeted loci in individual cells stably expressing the complementary fragments, enabling multiplexed visualization of endogenous proteins.

## Methods

**Molecular cloning**. The amino acid sequence of EBFP2 was obtained from a published report[16] and slightly altered (see Supplementary Data 2). EBFP2 was split at the same site at $GFP_{1-10/11}$. We introduced six substitutions to EBFP2$_{1-10}$ (S65T/Q80R/F99S/V128T/M153T/V163A), which altered the EBFP2$_{1-10/11}$ spectral property to Capri$_{1-10/11}$. EBFP2$_{1-10}$ and Capri$_{1-10}$ were synthesized (Integrated DNA Technologies) and cloned into the mammalian expression vector pcDNA3.1 (Invitrogen) between KpnI and EcoRI (NEB) by using an In-Fusion HD cloning kit (Takara Bio). In order to clone specific DNA fragments into a plasmid vector, we used the In-Fusion HD cloning system in this entire study.

For the expression of CFP$_{1-10}$ in mammalian cells, we introduced the corresponding point mutations into pcDNA3.1-GFP$_{1-10}$ (Addgene #70219) using Q5 High-Fidelity DNA Polymerase (NEB). Primers used were as follows: CFP$_{1-10}$-forward (5′-ACGCTTACGTggGGA GTTCAGTGC-3′), and CFP$_{1-10}$-reverse (5′-TGTTACGAGAGTCGGCCA-3′). To express Cerulean$_{1-10}$, we synthesized and cloned their DNA fragment into the KpnI/EcoRI sites of pcDNA3.1. For the nucleotide sequence of *Cerulean$_{1-10}$*, see Supplementary Data 1.

GFP permits circular permutation of the amino acid sequence[25]. By linking the N- and C-termini and cutting out a single β-strand, any one of the eleven β-strands would become a new split GFP-tag. Huang et al.[26] previously investigated all possible β-strands, and measured the solubility and relative reconstituted fluorescence intensity of each split GFP construct in *E. coli*. We tested Huang's design of the β-strands 7, 8, 9, or 10 system in human cells. We constructed plasmids encoding each of the β-strands fused to β-actin. We used mEmerald-Actin-C-18 (Addgene #53978) as the template for PCR amplification of *ACTB*. The *ACTB* gene was amplified using primers, in which DNA sequences encoding the β-strands were included in part (For the sequence information of the primers, see Supplementary Data 3). The resultant PCR products were cloned into the KpnI/EcoRI sites of pcDNA3.1.

*GFP$_{8-6}$, GFP$_{9-7}$, GFP$_{10-8}$*, and *GFP$_{11-9}$* were amplified from super-folder GFP OPT[1] by PCR and inserted into the KpnI/EcoRI sites of pcDNA3.1. For the nucleotide sequences of *GFP$_{8-6}$, GFP$_{9-7}$*, and *GFP$_{11-9}$*, see Supplementary Data 1.

To demonstrate the usefulness of mRuby4$_{11}$, sfCherry2$_{11}$, mNeonGreen2$_{11}$, and GFP$_8$, we labeled several cellular proteins with these split FP-tags. To construct plasmids of split FP-tag fusions, DNA fragments encoding cellular proteins were amplified by PCR with sets of primers (see also Supplementary Data 3) and cloned into the KpnI/EcoRI sites of pcDNA3.1. For the PCR amplification, we used the following DNA templates: mEmerald-Actin-C-18; mEmerald-Clathrin-15 (Addgene #54040); sfGFP-H2B-C-10 (Addgene #56367); pBABE-puro-GFP-wt-lamin A (Addgene #17662); mEmerald-Keratin-17 (Addgene #54134); sfGFP-Zyxin-6 (Addgene #56491).

The amino acid sequences of split FP-tags (i.e., GFP$_7$, GFP$_8$, GFP$_{10}$, GFP$_{11}$, mNeonGreen2$_{11}$, sfCherry2$_{11}$, and mRuby4$_{11}$) were listed in Supplementary Table 2.

sfCherry2$_{1-10}$, mNeonGreen2$_{1-10}$, and sfGFP-Zyxin plasmids[13] that can be transfected in mammalian cells are available through Addgene (#82603, #82610, and #56491, respectively).

**Mutagenesis and screening of libraries**. When engineering split orange-red FP variants, we adopted a complementation assay previously described to optimize split mCherry2 in *E. coli*[13]. We inserted a 30-aa spacer (GGGGSEGGGSGGPGSG GEGSAGGGSAGGGS) between the tenth and eleventh β strands of each of the following fluorescent proteins; mKO2, mRuby3, mApple, and mScarlet-I[15,21-23]. The corresponding DNA sequences were directly synthesized and then cloned into the BamHI/XhoI sites of the *E. coli* expression vector pET28a (Novagen).

The longer spacer insertion eliminated colony fluorescence of mKO2, mApple, and mScarlet-I, whereas colonies expressing spacer-inserted mRuby3 gave low signal (Supplementary Fig. 8). To improve the brightness of spacer-inserted mRuby3, we mutagenized it by using a GeneMorph II Random Mutagenesis Kit (Agilent). Mutants were expressed and inserted in pET28a. Plasmids were transformed into *E. cloni* EXPRESS Electrocompetent Cells (Lucigen). Transformation was performed by the Gene Pulser Electroporation Systems (BioRad). Colonies were grown on LB agar media (30 μg/mL Kanamycin) at 37 °C for 24 h and for additional 12–48 h at 37 °C after induction with 1 mM IPTG. For each round of mutagenesis, the number of colonies screened was at least 1 × 10$^4$. Colonies expressing spacer-inserted mRuby3 variants were screened for fluorescence with the ChemiDoc Imaging System (BioRad). The imaging system was equipped with an Epi-green 520–545 nm excitation source, a Green Epi 605/50 filter, and a cooled CCD camera.

Through library screening, we obtained an extremely bright variant of spacer-inserted mRuby3, which we named spacer-inserted mRuby4 (Supplementary Fig. 9). The *mRuby4$_{1-10}$* sequence of spacer-inserted mRuby4 was amplified by PCR and cloned into the KpnI/EcoRI sites of pcDNA3.1. For the nucleotide sequence of *mRuby4$_{1-10}$*, see Supplementary Data 1.

**Protein production and characterization of FPs**. For spectral characterization of FPs, we produced and purified recombinant proteins: full-length EBFP2, spacer-inserted EBFP2, full-length Capri, spacer-inserted mRuby3, spacer-inserted mRuby3, and spacer-inserted mRuby4 (the amino acid sequences of those were listed in Supplementary Data 2). We designed pET plasmids such that recombinant proteins were labeled at the C termini with poly-histidine tags. The plasmids were introduced into BL21(DE3) Competent *E. coli* cells (NEB) via transformation. Cells were grown in 250 mL LB medium at 37 °C for 6 h (OD$_{600}$ = 0.5), induced with IPTG (1 mM) for 4 h, and harvested by centrifugation. Cell pellets were lysed by French press. His-tagged proteins were purified with HisPur Cobalt Resin (Pierce). Proteins were further desalted into PBS pH7.4 using a GE Healthcare illustra NAP column (GE Healthcare). Extinction coefficients were calculated using Beer-Lambert law[10]. Quantum yields were determined using EBFP2[16], and Rhodamine B (Wako) as reference fluorophores. The absorbance signals of samples and reference were measured using a microreader (Biotek Synergy 2). Diluted samples

and reference were added into a quartz fluorescence cuvette (Thorlabs), and their integrated fluorescence intensities were measured by a fluorescence spectro-photometer (Hitachi F-7100). With the quantum yield of reference to be known, the final quantum yields of samples were attained using:

$$Qs = Qr \times (Ar/As) \times (Es/Er) \times (ns/nr)^2 \text{ [r and s refer to the reference and samples],}$$

where $Q$ is the quantum yield, $n$ is the refractive index, $A$ is the absorbance of solution, and $E$ is the integrated fluorescence intensity of emitted light.

**Fluorescence imaging**. Confocal microscopy images of mRuby4$_{1-10/11}$, GFP$_{1-10/11}$ and mNeonGreen2$_{1-10/11}$ were acquired on an inverted fluorescence microscope (Ti-E, Nikon) with a 100 × 1.45 NA oil immersion objective (Plan Apo, Nikon). The microscope was attached to the Dragonfly Spinning disk confocal unit (CR-DFLY-501, Andor). Two excitation lasers (40 mW 488 nm and 50 mW 561 nm lasers) were coupled to a multimode fiber passing through the Andor Borealis unit. A dichroic mirror (Dragonfly laser dichroic for 405-488-561-640) and band-pass filters (525/50 nm and 600/50 nm bandpass emission wheel filters) were selected for two-color imaging. The images were recorded with an EM-CCD camera (iXon, Andor).

Confocal microscopy images of EBFP2$_{1-10/11}$, Capri$_{1-10/11}$, CFP$_{1-10/11}$, and Cerulean$_{1-10/11}$, were collected by using an upright microscope (Axio imager Z2, Zeiss) with a 63 × 1.4 NA oil immersion objective (Plan Apo, Zeiss). The upright microscope had the LSM 880 Scan-head (Zeiss) with 32 channel GaAsP spectral PMT detector. It was equipped with six laser lines (Diode 405 nm; Argon 458, 488, 514 nm; HeNe 543, 633 nm). We used the 405-nm diode laser for EBFP2$_{1-10/11}$, and Capri$_{1-10/11}$ (main beam filter MBS-405 and 488/543, 409–491 nm barrier filter), the 458-nm Argon line for CFP$_{1-10/11}$, and Cerulean$_{1-10/11}$ (main beam filter MBS-458/514, 454–518 nm barrier filter). To characterize a spectrum from each individual FP, spectral images were acquired. We used 8.9 nm channel widths, meaning that each of the 32 channels on the spectral detector captured light over 8.9 nm bandwidth of the visible and near infrared spectrum. Sequential spectral image acquisitions were achieved in order of ascending wavelength of the excitation laser: 405, 488, and 543 nm. A typical data set consisted of 32 images (1024 × 1024 pixels), corresponding to different wavelengths from 410 to 695 nm. We then performed linear unmixing (Figs. 2h, 4g, and Supplementary Figs. 12–14) using Carl Zeiss' Zen software.

Confocal microcopy images showed average intensity z projections, unless otherwise noted in the figure legends. Analysis of the confocal images was performed on Fiji software (NIH).

**Flow cytometry**. Fluorescence of cells was measured using CytoFLEX (Beckman Coulter) in the CTEGD Cytometry Shared Resource at the University of Georgia (UGA). The instrument had four excitation lasers (405, 488, 561, and 610 nm) and three band-pass filters (450/45, 525/40, and 610/20 nm). Post-acquisition analysis was carried out using FlowJo software (Treestar, Inc).

**Cell culture, transfection, and drug treatment**. HEK 293, HEK 293T, HeLa, and U2OS cells (gifted by Drs. Eggenschwiler and Kipreos, UGA) were grown in Dullbecco's Eagle's medium (HyClone), supplemented with fetal bovine serum (10%, v/v; Atlanta Biologicals) and penicillin/streptomycin (100 U/mL penicillin and 100 μg/mL streptomycin; HyClone). Cells were cultured at 37 °C and 5% CO$_2$ in a humidified incubator. Plasmids were transfected at 400–800 ng DNA per well with Lipofectamine 2000 (3 μL, Invitrogen) or polyethylenimine (3 μL of 1 mg/mL PEI, Polysciences, Inc.) into Nunc Lab-Tek II Chambered Cover Glass (size: 8 wells, Nalge Nunc International) or Corning Costar Cell Culture Plates (size: 12 or 24 wells, Corning). In particular, for the quantitative comparisons in cellular fluorescence intensity (Fig. 4a and Supplementary Figs. 4 and 6), we transfected the plasmids of FP-tags and their partners at 400 ng and 800 ng, respectively. Cells were fixed with buffered 4% paraformaldehyde (Electron Microscope Sciences), mounted with PBS, and imaged by confocal microscopy.

For spectral imaging of multicolor H2B fusions (Fig. 2h and Supplementary Figs. 13 and 14), we used a Cdk1 inhibitor (10 μM of RO-3306, Sigma-Aldrich) to synchronize HEK 293 cells. HEK 293 cells were treated with the inhibitor for 18 h, blocked in the G2/M phase. For release from the inhibitor, we washed the culture five times with prewarmed culture media. Released cells returned to normal cell cycle progression, and were eventually fixed with 100% ice-cold methanol and mounted with PBS for microscopy.

**Knock-in cell creation**. For knock-in of mRuby4$_{11}$ into the *HIST2H2BE* locus, we ordered 200-nt HDR templates in single-stranded DNA (5′-gccccggcgagctggccaagca cgccgtgtccgagggcaccaaggcggtcaccaagtacaccagctccaagGGTGGCGGCGAAACCTAC GTAGTGCAAAGAGAAGTGGCAGTTGCCAAATACAGCAACtgagtccctgccggga cctggccgctcgctcgctcgagtcgccggctgcttgactccaaaggctcttttcagag-3′, Integrated DNA Technologies). Cas9 protein was expressed in *E. coli* and purified by the Kipreos laboratory at UGA as described previously[11]. sgRNA and Cas9/sgRNA ribonucleoprotein complexes were prepared as described before[12]. After the treatment of HEK293 FT cells with nocodazole (200 ng/mL, Sigma-Aldrich) for 16 h, we performed electroporation on an Amaxa Nucleofector 2b device with Nucleofector Solution V reagents (Lonza).

Nocodazole-treated cells were resuspended at a concentration of $1 \times 10^4$ cells/μL in 100 μL of Nucleofector Solution V. We added cells to the RNP/donor template mixture (50 μL), electroporated using the Q-001 program, and quickly transferred to 12-well plates with pre-warmed media. Electroporated cells were cultured for 2–5 days and transfected with mRuby4$_{1-10}$ plasmid.

**Statistics and reproducibility**. All experiments for the measurement of signal levels were replicated multiple times independently. Statistical analyses were performed using GraphPad Prism 7. Error bars in all figures refer to the standard error of the mean.

**Reporting summary**. Further information on research design is available in the Nature Research Reporting Summary linked to this article.

## Data availability

Relevant plasmids and sequences have been deposited in Addgene (www.addgene.org). The raw data referring to the plots shown in the main figures are provided in Supplementary figures. All relevant data are available from authors upon requests.

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

## Acknowledgements

We thank Zahra Abdul Nawaz and Dr. Edward Kipreos (University of Georgia) for assistance in in vitro Cas9 expression and purification, Dr. Rick Tarleton (University of Georgia) for use of his Nucleofector equipment, and members of the Kamiyama Lab for comments on the manuscript. We also acknowledge the assistance of the Biomedical Microscopy Core and the CTEGD Cytometry Shared Resource Laboratory. This work was supported by the University of Georgia Faculty Seed Grant (to D.K.) and an NIH R01 NS107558 (to R.T. and D.K.). R.T. was supported by a predoctoral fellowship from the Nakajima Foundation.

## Author contributions

R.T. and D.K. conceived and designed the experiments. R.T. performed random mutagenesis, protein labeling, CRISPR-mediated knock-in, flow cytometry, and imaging experiment. R.T. analyzed the data. F.J., R.T., and J.X. performed in vitro characterizations of split FPs. D.K. wrote the manuscript.

## Competing interests

The authors declare no competing interests.
