## [Peer Review File · Communications Biology]

Reviewers' comments:

Reviewer #1 (Remarks to the Author):

The authors engineered a palette of split fluorescent proteins including: blue fluorescent and bright Capri(1-10/11), red fluorescent mRuby4(1-10/11), and 3 new split GFP with different split sites including 8-6/7; 9-7/8; 11-9/10. These are important tools for labeling endogenous proteins. I recommend publication of this manuscript, with only a few minor comments.

1. In the introduction, the authors may want mention directly what they have engineered, including split mRuby4, Capri.
2. The authors may want to reduce the introduction of sfCherry2 and expand the experimental details on split EBFP2, e.g. how it was engineered? Using long linker?
3. Curious why didn't the authors engineer a split BFP starting from the already engineered split GFP plus Y66H mutation? Y66H is the key mutation previously characterized to blue shift the GFP spectra. This might have been a faster way than the author's current strategy by splitting EBFP2, followed by directed evolution for a split EBFP2, and then borrowing mutations from split GFP. Maybe the authors have other reasons?

Reviewer #2 (Remarks to the Author):

In this paper, the authors expand a split fluorescent protein system to multi-color imaging (seven-colors). The paper is generally well written (but too short for abstract and introduction), and the quality of data is sufficient.

But, the approach is not new, and the advantage of split FPs developed by the authors is an insufficient explanation. Some papers, including those by the authors, have already reported different colors of split FPs, such as cyan, yellow, and red. So, the authors should introduce the split FPs reported so far in introduction and describe the advantage and utility of newly developed split FPs against them. Moreover, this is a critical point, the authors should show the utility and innovative advantage of developed split FPs in probing questions about cellular events or proteins function.

I have some minor comments to be addressed in detail below:

Minor concerns:

- 1) In Fig. 2b, Multi-color split FPs could distinguish each cell by spectral imaging. Although the authors said that "Altogether, these experiments illustrate seven-color spectral imaging of cellular proteins.", could we perform this seven-color spectral imaging for different proteins in a cell?
- 2) In lines 137-140, I don't understand the utility of circular permutation for split FPs because the fluorescent intensity of all circular permuted split GFPs are decreased. I think the direct evolution of original split FPs has the potential to increase fluorescent intensities.
- 3) In lines 213-214, please provide culture conditions such as temperature and time.

4) In supplemental figures, there are a mix of standard deviations and SEM for error bars, so please unify throughout the experiments.

Reviewer #3 (Remarks to the Author):

The manuscript from Tamura and Kamiyama describes the evolution of spectrally distinct split-fluorescent proteins and provides some biological imaging applications, including dual-color and multispectral confocal imaging of cells. While some aspects of the article are new and of interest, such as the successful development of a split-mRuby4 or the spectral imaging application, the overall lack of characterization of

the different FP1-10, significantly dampens the enthusiasm for the paper. Indeed, the manuscript describes rather incremental progresses compared to previous works with similar split-fluorescent proteins (Cabantous, Bystroff, Huang, Pinaud, Waldo) and, as written, provides limited new insights with regard to FP1-10 properties or applications. For instance, it is not clear from the manuscript why multi-color imaging with seven FP1-10 is more interesting/useful than multi-color imaging with seven full-length FPs. The manuscript would be greatly improved if the authors characterize each FP1-10-/FP-11 pair in vitro, such that they can become useful tools for biologists.

Major comments:

1- There appear to be differences in staining patterns for actin and actin stress fibers across the various split-fluorescent proteins tested in HeLa cells (Fig. 1a-g and Supplementary Fig. 3). Some split-fluorescent protein tags display surprisingly thick actin stress fibers (e.g. Cerulean1-10), while other display thin and barely visible stress fibers (e.g. Ypet1-10, CFP1-10, YFP1-10). In the case of EBFP21-10, significant perinuclear staining is also observed. Overall the staining patterns appear different from those previously reported by the corresponding author (Nat. Comm. 2016, 7, #11046) and by Feng et al. (Nat. Comm. 2017, 8, #370) using the same β -actin fusion to GFP11 or sfCherry211. Since GFP11-actin is common to all the FP1-10 tested, what difference between the FP1-10 could explain such wide variations in the staining of actin structures? The author should also compare the split-fluorescent protein staining in HeLa cells with the corresponding full-length fluorescent protein staining (e.g. β -actin fusions to EBFP2, Capri, CFP, Cerulean, Ypet and YFP).

2- Throughout the manuscript, the authors mention that they compare brightness between different split-fluorescent protein (lines 41-42, 44-45, 48-49, 63-64, 72-75, 122-123, 195, 205). The authors are certainly aware that a fluorophore brightness is defined as the product of its quantum yield and of its extinction coefficient. It should not be confused with fluorescence intensity, which, in the case of FPs expressed in cells, comprises FP brightness, FP expression levels, FP folding efficiency, chromophore maturation and FP stability. In the case of split-fluorescent protein tags, cellular fluorescence intensity depends on additional parameters such as bimolecular complementation between fragments and oligomeric state of the FP1-10-tag, some of which were shown to be dimeric depending on concentrations/expression levels (see for instance Koker et al. Sci. Rep. 2018, 8, #5344).

In the manuscript, the authors do not systematically control for cell expression levels of the FP1-10-tags (except in Supplementary Figures 2 and 7) nor for those of the FP11 fusions (except in the case of mRuby411 CRISPR, but see also point #3, below). In addition, they do not provide data on the intrinsic properties of each FP1-10/FP11 pairs (e.g. brightness, bimolecular complementation, etc...). It is therefore difficult to conclude if the increased cell fluorescence intensities for some FP1-10 compared to others actually stem from improved photophysics (e.g. brightness) or from other parameters, such as increased expression (of one or both fragments), better folding, better stability or

better complementation. While I understand that controlling cell expression levels for both FP1-10 and FP11-fusion is somewhat of a difficult task, it is critical that the authors provide at least some in vitro characterization of the molecular brightness for each split-fluorescent protein.

3- The authors co-transfect FP1-10 and FP11-fusions from two separate pcDNA3 plasmids that nonetheless have the same CMV promoter. This can often lead to issues of homologous promoter interferences where competition for limited amount of general transcription factors might lead to anti-correlated transcriptions of the FP1-10 tags and of the FP11-fusions (See for instance: Huliak et al. *DNA and Cell Biology* 2012, 31 (11), p: 1580–1584). Could the author please comment on how this might affect the quantitative comparisons in cell fluorescence intensity described throughout the manuscript?

Minor comments:

4- In supplementary Figure 4, the authors report quantifications from mean fluorescence intensity projections for a limited region of 20 μm on each cell. Why was the mean value used as opposed to the sum fluorescence intensity projection, which is less prone to bias? The authors should provide a total fluorescence intensity quantification similar to what is generally measured by FACS. Also, does the 20 μm region chosen cover each cell entirely? Was a background correction applied? Are the differences between CFP1-10 and Cerulean1-10 and between YFP1-10 and YPet1-10 significant?

5- In Figure 2A, it would be very useful if the authors could additionally provide the absorption spectrum for each FP1-10.

We would like to thank all the reviewers for their comments and feedbacks. We have revised our manuscript according to their suggestions, which greatly improved the transparency of this manuscript. Please see our responses to the comments in blue text.

Reviewers' comments:

Reviewer #1 (Remarks to the Author):

The authors engineered a palette of split fluorescent proteins including: blue fluorescent and bright Capri(1-10/11), red fluorescent mRuby4(1-10/11), and 3 new split GFP with different split sites including 8-6/7; 9-7/8; 11-9/10. These are important tools for labeling endogenous proteins. I recommend publication of this manuscript, with only a few minor comments.

We thank the reviewer for the supportive comments.

1. In the introduction, the authors may want mention directly what they have engineered, including split mRuby4, Capri.

Thank you for your suggestion. We have improved the clarity of this manuscript by adding the information of those split FPs into the Introduction section (page 4, lines 42-53).

We have revised the Introduction section to make this point more straightforward. We listed what we created (i.e., EBFP2_{1-10/11}, Capri_{1-10/11}, mRuby4₁₋₁₀, and circularly permuted GFP fragments), and also briefly described our strategies for the engineering of EBFP2_{1-10/11}, Capri_{1-10/11}, and mRuby4₁₋₁₀.

2. The authors may want to reduce the introduction of sfCherry2 and expand the experimental details on split EBFP2, e.g. how it was engineered? Using long linker?

We have rewritten this part of manuscript on page 5 lines 56-70 to make the experiment it more explicit.

A previous report suggested that six substitutions (N40I/T106K/E112V/K166T/I167V/S206T) could improve the complementation efficiency of split-GFP (Cabantous et al., *Nat Biotechnol* 23(1), 102-7 (2005)). Based on this report we applied the same substitutions to the large fragment of EBFP2 (i.e. EBFP2₁₋₁₀), which was split from the same site as GFP_{1-10/11}. However, the brightness of the reconstituted split EBFP2 was noticeably weaker than full-length EBFP2. In order to improve the brightness of it, we utilized six more substitutions to EBFP2₁₋₁₀ that have been shown to favor the stability and folding rate of GFP (Pedelacq et al., *Nat Biotechnol* 24(1), 79-88 (2006)). The resulting split FP was a blue-colored variant termed split Capri.

3. Curious why didn't the authors engineer a split BFP starting from the already engineered split GFP plus Y66H mutation? Y66H is the key mutation previously characterized to blue shift the GFP spectra. This might have been a faster way than the author's current strategy by splitting EBFP2, followed by directed evolution for a split EBFP2, and then borrowing mutations from split GFP. Maybe the authors have other reasons?

Yes, the reviewer is correct in pointing out the use of the Y66H mutation. We also considered using GFP-Y66H (known as EBFP); however, previous literature reported that EBFP2 is 4-fold brighter and > 500-fold more photostable than EBFP, which motivated us to start engineering the split FP from EBFP2 (Ai et al., *Biochemistry* 46(20), 5904-10, (2007)).

Reviewer #2 (Remarks to the Author):

In this paper, the authors expand a split fluorescent protein system to multi-color imaging (seven-colors). The paper is generally well written (but too short for abstract and introduction), and the quality of data is sufficient.

Thanks a lot for the encouraging comments for the reviewer!

But, the approach is not new, and the advantage of split FPs developed by the authors is an insufficient explanation. Some papers, including those by the authors, have already reported different colors of split FPs, such as cyan, yellow, and red. So, the authors should introduce the split FPs reported so far in introduction and describe the advantage and utility of newly developed split FPs against them.

In the revised manuscript, we have included additional detailed information of split FPs that have been already reported (such as split CFP, YFP, and sfCherry).

While split-FPs have become a versatile tool used for different imaging applications, labeling multiple proteins within a single cell remains a challenge. Nevertheless, recent advancements have made sfCherry_{1-10/11} system into a new orthogonal split FP tag in addition to the existing split GFP system, which allows two color imaging (Feng et al., *Nat Commun*, 8(1), 370, (2017)). However, the lack of orthogonal split FP tags with distinct fluorophores limits the number of proteins that can be tagged in a single cell simultaneously. In this study, we engineered new split FP color variants (i.e. EBFP2_{1-10/11}, Capri_{1-10/11}, and mRuby4₁₋₁₀) and illustrated the potential utility of circular permutation for diversifying orthogonal split FPs. As a result, this approach overcomes the multiplexing limitation of split FP tagging. We have added a summary description on the advantage and utility of new split FPs in the second and third paragraphs of the Introduction section (page 3 lines 33-41).

Moreover, this is a critical point, the authors should show the utility and innovative advantage of developed split FPs in probing questions about cellular events or proteins function.

In this study we demonstrate the utility of the new split FPs by conducting multicolor split FP imaging in single cells. We over-expressed the four orthogonal split FPs (Capri_{1-10/11}, GFP_{9-7/8}, mNeonGreen2_{1-10/11}, and mRuby4_{1-10/11}) as fusions to H2B, Lamin A/C, β -actin, and Zyxin. Consequently, we observed the correct localizations of these proteins (new Fig. 4g). Additionally, for some cells, zyxin was localized to the nucleus while the majority of the protein was localized to the focal adhesions (new Supplementary Fig. 18). These findings are also supported by similar observations in a different cell line (Fujita et al., *BMC Cell Biol*, 10, 6 (2009)). As a large protein that lacks a nuclear localization signal, zyxin must enter the nucleus with other NLS-containing proteins. Altogether, here we demonstrate the utility of multiplexed split-FP labeling in single cells. This new piece of data has been added on page 10, lines 183-197.

I have some minor comments to be addressed in detail below:

Minor concerns:

1) In Fig. 2b, Multi-color split FPs could distinguish each cell by spectral imaging. Although the

authors said that "Altogether, these experiments illustrate seven-color spectral imaging of cellular proteins.", could we perform this seven-color spectral imaging for different proteins in a cell?

Following this suggestion, we have performed additional experiments (please see our response to the major comments). With available orthogonal split FPs, we have performed four-color spectral imaging for distinct proteins in single cells (Capri_{1-10/11}, GFP_{9-7/8}, mNeonGreen2_{1-10/11}, and mRuby4_{1-10/11}). In order to increase the number of colors, we need to engineer more orthogonal pairs using circularly permuted fragments of other FPs (e.g., EBFP2 and sfCherry2). As much as we would love to have additional orthogonal pairs, we do not think it is practical within a reasonable time frame to revise this manuscript.

2) In lines 137-140, I don't understand the utility of circular permutation for split FPs because the fluorescent intensity of all circularly permuted split GFPs are decreased. I think the direct evolution of original split FPs has the potential to increase fluorescent intensities.

It is likely that our writing has generated some confusion or misunderstanding for the reviewer. The purpose of circularly permutation is to provide more orthogonal split FP systems. In our manuscript, we studied the binding specificities of GFP_{8-6/7}, GFP_{9-7/8}, GFP_{11-9/10}, and GFP_{1-10/11}. These split GFP pairs are orthogonal to GFP_{1-10/11}. Such an orthogonal interaction was validated by multiplexed imaging in U2OS cells (new Fig. 4f-g).

To avoid such confusions, we have modified the text of the Introduction. In addressing the major comments of this reviewer, we have carefully explained the utility of orthogonal split FP systems. See revised text on page 3 lines 33-41.

3) In lines 213-214, please provide culture conditions such as temperature and time.

We have added information regarding the culture conditions (page 15, line 289-291).

4) In supplemental figures, there are a mix of standard deviations and SEM for error bars, so please unify throughout the experiments.

We have revised the manuscript according to the suggestion.

Reviewer #3 (Remarks to the Author):

The manuscript from Tamura and Kamiyama describes the evolution of spectrally distinct split-fluorescent proteins and provides some biological imaging applications, including dual-color and multispectral confocal imaging of cells. While some aspects of the article are new and of interest, such as the successful development of a split-mRuby4 or the spectral imaging application, the overall lack of characterization of the different FP1-10, significantly dampens the enthusiasm for the paper. Indeed, the manuscript describes rather incremental progresses compared to previous works with similar split-fluorescent proteins (Cabantous, Bystroff, Huang, Pinaud, Waldo) and, as written, provides limited new insights with regard to FP1-10 properties or applications. For instance, it is not clear from the manuscript why multi-color imaging with seven FP1-10 is more interesting/useful than multi-color imaging with seven full-length FPs. The manuscript would be greatly improved if the authors characterize each FP1-10-/FP11 pair in vitro, such that they can become useful tools for biologists.

Thank you for your comments. One of the prominent applications of split FPs is the generation of human cells with fluorescently tagged endogenous proteins via CRISPR/Cas9-mediated homology-directed repair (new Supplementary Fig. 11). As the split FP tag is quite small (~1/10 of the size of full-length FP), a simpler cloning-free knock-in approach can be carried out with a short donor oligo (Leonetti et al. *PNAS*, 113(25), 3501-8, (2016)). Additionally, short inserts can be introduced to the host genome at high recombination efficiencies (Paix et al. *PNAS*, 114(50), 10745-10754, (2017)). At last, the simplicity and efficiency of this approach aids the generation of multiple split FP tag insertions into single cells for future experiments. We have included this discussion into the main text, please see page 11 lines 206-216.

Major comments:

1- There appear to be differences in staining patterns for actin and actin stress fibers across the various split-fluorescent proteins tested in HeLa cells (Fig. 1a-g and Supplementary Fig. 3). Some split-fluorescent protein tags display surprisingly thick actin stress fibers (e.g. Cerulean1-10), while other display thin and barely visible stress fibers (e.g. Ypet1-10, CFP1-10, YFP1-10). In the case of EBFP21-10, significant perinuclear staining is also observed. Overall the staining patterns appear different from those previously reported by the corresponding author (Nat. Comm. 2016, 7, #11046) and by Feng et al. (Nat. Comm. 2017, 8, #370) using the same β -actin fusion to GFP11 or sfCherry211. Since GFP11-actin is common to all the FP1-10 tested, what difference between the FP1-10 could explain such wide variations in the staining of actin structures? The author should also compare the split-fluorescent protein staining in HeLa cells with the corresponding full-length fluorescent protein staining (e.g. β -actin fusions to EBFP2, Capri, CFP, Cerulean, Ypet and YFP).

In the revision, we have included additional detailed information (such as Supplementary Figs 2 and 7) and improved the explanation in the main text.

split Cerulean (see page 6 lines 95-102)

Based on this comment, we have checked whether the thickness of actin stress fibers is abnormal in the obtained image of split Cerulean. Indeed, there is a significant difference between expressing split Cerulean vs. staining with Rhodamine-Phalloidin in HeLa cells (Phalloidin is used as a standard comparison to split FPs). However, HeLa cells expressing a full-length Cerulean fusion have not displayed abnormal thickening of the bundles (new Supplementary Fig. 7). This type of artifact is commonly seen in oligomeric FPs when these FPs are targeted to two-dimensional structures (Cranfill et al. *Nat Methods*, 13(7), 557-62 (2016)). Therefore, we speculate that split Cerulean (but not full-length Cerulean) could be oligomeric which we plan to validate in split Cerulean with various conditions. Although our observation raises a concern about the oligomeric potential of split Cerulean, for some applications such as CFP Reconstitution Across Synaptic Partners (CRASP; Li et al., *Biochem Biophys Res Commun*, 469(3), 352-6, (2016)), monomeric proteins are not required; thus, this bright CFP variant can be used alternatively.

split CFP and YFP (see page 6 lines 84-86)

We agree with the reviewer's point. In particular, the overall brightness of split CFP is relatively weak, making it difficult to visualize thin actin filaments. Contrary to our perception, there were not any statistically significant differences in the thickness of actin stress fibers for split CFP, split YFP or Rhodamine-Phalloidin.

split EBFP2 (see page 5 lines 65-67)

When used to tag β -actin in cells, the split EBFP2 reconstitutes to give fluorescent signals. However, the signals are found to be extremely weak, which renders it a challenge to avoid the increased autofluorescence exhibited in this spectral region in some cases (e.g. the actin image). Specifically, the increased autofluorescence background has been observed in the perinuclear region of un-transfected HeLa cells when using UV light (new Supplementary Fig. 2) which is also consistent with a perinuclear signal observed in Fig. 1a.

Quantification of actin bundle thickness shown in the figures:
The mean values for each actin stress fiber were calculated. $n = 10-13$ actin fibers per cell. * $P < 0.01$ (two-tailed t-test).

2- Throughout the manuscript, the authors mention that they compare brightness between different split-fluorescent protein (lines 41-42, 44-45, 48-49, 63-64, 72-75, 122-123, 195, 205). The authors are certainly aware that a fluorophore brightness is defined as the product of its quantum yield and of its extinction coefficient. It should not be confused with fluorescence intensity, which, in the case of FPs expressed in cells, comprises FP brightness, FP expression levels, FP folding efficiency, chromophore maturation and FP stability. In the case of split-fluorescent protein tags, cellular fluorescence intensity depends on additional parameters such as bimolecular complementation between fragments and oligomeric state of the FP1-10-tag, some of which were shown to be dimeric depending on concentrations/expression levels (see for instance Koker et al. Sci. Rep. 2018, 8, #5344).

In the manuscript, the authors do not systematically provide cell expression levels of the FP1-10-tags (except in Supplementary Figures 2 and 7) nor for those of the FP11 fusions (except in the case of mRuby411 CRISPR, but see also point #3, below). In addition, they do not provide data on the intrinsic properties of each FP1-10/FP11 pairs (e.g. brightness, bimolecular complementation, etc...). It is therefore difficult to conclude if the increased cell fluorescence intensities for some FP1-10 compared to others actually stem from improved photophysics (e.g. brightness) or from other parameters, such as increased expression (of one or both fragments), better folding, better stability or better complementation. While I understand that controlling cell expression levels for both FP1-10 and FP11-fusion is somewhat of a difficult task, it is critical that the authors provide at least some *in vitro* characterization of the molecular brightness for each split-fluorescent protein.

We fully agree with the reviewer on the practical importance of *in vitro* characterization and thank the reviewer for the suggestion. In the revised manuscript, we have measured the extinction coefficient (EC) and quantum yield (QY) values of full-length EBFP2, split EBFP2,

split Capri, full-length mRuby3, split mRuby3, and split mRuby4 (see also the new Supplementary Table 1).

split BFP variants (see page 5 lines 70-74)

split Capri shows a peak extinction coefficient of $37 \times 10^3 \text{ M}^{-1}\text{cm}^{-1}$ and quantum yield of 0.13, which surpass the EC and QY of split EBFP2 ($23 \times 10^3 \text{ M}^{-1}\text{cm}^{-1}$ and 0.07, respectively), demonstrating that split Capri has superior molecular brightness to split EBFP2.

split Ruby variants (see page 7 lines 115-119)

We have also measured the EC and QY of split mRuby3 and split mRuby4. split mRuby4 has a higher EC and increased QY when compared to split mRuby3 ($32 \times 10^3 \text{ M}^{-1}\text{cm}^{-1}$ and 0.23 for split mRuby3; $93 \times 10^3 \text{ M}^{-1}\text{cm}^{-1}$ and 0.32 for split mRuby4). Our library screening has indeed resulted in the creation of a brighter split Ruby variant.

We hope that these characterizations will provide useful information to guide the practical application of these newly engineered split FPs.

3- The authors co-transfect FP1-10 and FP11-fusions from two separate pcDNA3 plasmids that nonetheless have the same CMV promoter. This can often lead to issues of homologous promoter interferences where competition for limited amount of general transcription factors might lead to anti-correlated transcriptions of the FP1-10 tags and of the FP11-fusions (See for instance: Huliak et al. DNA and Cell Biology 2012, 31 (11), p: 1580–1584). Could the author please comment on how this might affect the quantitative comparisons in cell fluorescence intensity described throughout the manuscript?

We agree with the reviewer that co-transfection of multiple constructs would result in slight decrease in the FP₁₋₁₀ and FP₁₁ production. Additionally, Huliak *et al.* reported that many factors including types of promoters, amounts of DNA used for transfection, and the nature of host cells can result in interference. Therefore, we transfected HEK 293T cells with two pcDNA3 plasmids and maintain the same amount of DNA in each experiment for comparison.

In Supplementary Figs 4 and 6, pcDNA3-FP₁₁ and pcDNA3-mScarlet-FP₁₋₁₀ were co-transfected at 400 ng and 800 ng DNA, respectively, per well with 3 μL of Lipofectamine-2000. In addition, we have used the signal of mScarlet to normalize the differences of gene expression levels. We have added clarification to the Method section regarding the amounts of DNA in these figures. Please see page 17, lines 344-346.

In Supplementary Fig. 10, we have performed a cellular fluorescence measurement and compared the signal of spacer-inserted mRuby4 to full-length mRuby3. pcDNA3-TagBFP-spacer-inserted mRuby4 and pcDNA3-TagBFP-mRuby3 were individually transfected at 800 ng DNA (TagBFP is to normalize gene expression levels); therefore, such an interference is not an issue.

Minor comments:

4- In supplementary Figure 4, the authors report quantifications from mean fluorescence intensity projections for a limited region of 20 μm on each cell. Why was the mean value used as opposed to the sum fluorescence intensity projection, which is less probe to bias? The authors should provide a total fluorescence intensity quantification similar to what is generally measured by FACS. Also, does the 20 μm region chosen cover each cell entirely? Was a background correction applied? Are the difference between CFP1-10 and Cerulean1-10 and

between YFP1-10 and YPet1-10 significant?

Thanks for the suggestion. We have performed a total fluorescence intensity quantification and revised the figure (new Supplementary Fig. 6). A 20- μm diameter region covers the nucleus; Note that we have expressed GFP₁₁-H2B with Cerulean₁₋₁₀ or CFP₁₋₁₀. A background correction had been applied.

Yes, we found that Cerulean₁₋₁₀ signal was significantly brighter than that of CFP₁₋₁₀. On the other hand, there was no statistically significant difference between YFP₁₋₁₀ and YPet₁₋₁₀. Therefore, we have decided to remove the results regarding YPet_{1-10/11} from the revised manuscript. We hope this change will allow the attention of the reader to be focused on the improvement of Cerulean_{1-10/11} over CFP_{1-10/11}.

5- In Figure 2A, it would be very useful if the authors could additionally provide the absorption spectrum for each FP1-10.

We have included the absorption spectra for split EBFP2, Capri, and mRuby4 in the new Supplementary Fig 3.

Other changes:

To fit the manuscript to the Communications Biology format, we have split two figures into four figures and also added the Discussion section.

Reviewers' comments:

Reviewer #2 (Remarks to the Author):

In the revised manuscript, the authors have improved their manuscript according to our comments. However, I still have some questions in some points.

1)

The authors should indicate changes in red in the text for clarity.

2)

page 3 lines 33-41

The authors stated that "Although these self-complementing split GFP variants have already become a powerful and

34 versatile tool for various microscopy applications".

To better understanding the advantage of split FPs over full-length, the authors should explain the specific applications of "various microscopy applications" here.

3)

Supplementary Figure 4

Please explain the abbreviation of SEM.

Why did the author use standard deviation in the previous manuscript and standard error of the mean in the revised manuscript?

4)

There is little discussion and a lot of redundancy in the Discussion. I suggest to the authors that they may consider to combine Results and Discussion.

Reviewer #3 (Remarks to the Author):

Except for the new results on the nuclear accumulation of zyxin, I am relatively satisfied with the answers provided by the authors in response to comments made by the three reviewers.

1-I recommend that extinction coefficient values in Supplementary Table 1 be provided in standard units of $M^{-1}cm^{-1}$, and that calculation of brightness be appropriately corrected.

2-The results provided on YFP1-10 in the current version of the manuscript have now very limited interests. Indeed, comparisons to YPet has been removed by the authors and YFP1-10 data are not being discussed. I recommend that the authors remove them to allow the attention of the readers to be focused on the improvement of Cerulean1-10/11 over CFP1-10/11.

3-There appears to be a YPet image overlaid in FOV#3 in Supp Fig. 14 It should be removed.

4-While I appreciate the authors' efforts to demonstrate some biological applications of their split-FPs in response to reviewer 2 and to my own comments, I find the data on the nuclear localization of zyxin to be of limited interest, and not representative of the advantages that split-FP probes might provide.

For instance, it is not clear to me why such an observation could not have been made with traditional full FP fusions. Have the author also studied the nuclear localization of zyxin with full FP fusions? If so, why such an accumulation would be "revealed" by split-FP and not full FPs? The authors need to be much clearer on these points and they should provide some explanation. From a quick survey of the literature, it appears that the accumulation of zyxin in the nucleus has been observed by many other groups (see ref. 23 cited by the authors and other references therein). Specifically, in the case of ref. 23, such accumulation depends on the inhibition of CRM1-mediated nuclear exportation by leptomycin B. Why would tagging zyxin with split-FPs induce a similar nuclear accumulation? Does split-FP tagging prevent a normal function of nuclear export signals within zyxin, thus phenocopying the effect of leptomycin? In the absence of further explanation and discussion, these results have, unfortunately, very little interest and fail at conveying the advantages of using split-FP fusions for biological imaging. In fact, as currently described, the zyxin data raise more questions on the use of split-FPs for protein tagging than they provide answers.

Please see the point-by-point answers to the reviewers' comments below in blue color.

Reviewers' comments:

Reviewer #2 (Remarks to the Author):

In the revised manuscript, the authors have improved their manuscript according to our comments. However, I still have some questions in some points.

We greatly appreciate the positive comments from the reviewer.

1)

The authors should indicate changes in red in the text for clarity.

In this revised manuscript, we have shown the changes in red.

2)

page 3 lines 33-41

The authors stated that "Although these self-complementing split GFP variants have already become a powerful and [34] versatile tool for various microscopy applications". To better understanding the advantage of split FPs over full-length, the authors should explain the specific applications of "various microscopy applications" here.

We have revised the Introduction and edited the second paragraph so that the specific applications are discussed:

(Introduction, 2nd paragraph): "In particular, the short fragment of GFP₁₁ can be efficiently introduced at the genomic locus of the gene of interest without perturbing local genomic structure, allowing us to reliably produce endogenously tagged cell lines (Kamiyama et al., *Nat Commun* 7, 1046 (2016); Paix et al., *Proc Natl Acad Sci USA* 114(50), 10745-10754 (2017)). Additionally, we have successfully generated a library of human cells with GFP₁₁-tagged endogenous proteins via CRISPR/Cas9-mediated homology-directed repair (HDR) and demonstrated that GFP₁₁-tag is compatible with a wide range of cellular proteins including enzymes, receptors, transport proteins, and structural proteins (Leonetti et al., *Proc Natl Acad Sci USA* 113(25), 3501-8 (2016))"

3)

Supplementary Figure 4

Please explain the abbreviation of SEM.

Why did the author use standard deviation in the previous manuscript and standard error of the mean in the revised manuscript?

It has been brought to our attention that there is a mix of standard deviation and SEM (standard error of the mean) for error bars in the previous manuscript, so we have unified them into SEM in the revised manuscript.

In figures (including Supplementary Fig. 4), we have used the mean fluorescence intensity, relative to the original cases of split FPs, to examine whether mutating FP₁₋₁₀ fragments (such as Capri₁₋₁₀, Cerulean₁₋₁₀, and mRuby₄₋₁₀) enhances the signal. Accordingly, we have compared their relative mean values to the new variants. In these cases, reporting variability in

measurements on different cells within the same experiment would be more appropriate. We therefore chose SEM to show their associated uncertainty.

4)

There is little discussion and a lot of redundancy in the Discussion. I suggest to the authors that they may consider to combine Results and Discussion.

Thank you for your suggestion. The paragraph in the Discussion has been re-written and combined with the Results section. See the revised text on pages 10-11 lines 201-213.

Reviewer #3 (Remarks to the Author):

Except for the new results on the nuclear accumulation of zyxin, I am relatively satisfied with the answers provided by the authors in response to comments made by the three reviewers.

We thank the reviewer for the positive comments.

1-I recommend that extinction coefficient values in Supplementary Table 1 be provided in standard units of $M^{-1}cm^{-1}$, and that calculation of brightness be appropriately corrected.

Following this suggestion, we have changed the units of EC to $M^{-1}cm^{-1}$ in Supplementary Table 1.

2-The results provided on YFP1-10 in the current version of the manuscript have now very limited interests. Indeed, comparisons to YPet has been removed by the authors and YFP1-10 data are not being discussed. I recommend that the authors remove them to allow the attention of the readers to be focused on the improvement of Cerulean1-10/11 over CFP1-10/11.

Thanks for the reviewer's feedback. We have revised the manuscript according to the suggestion (page 6, lines 86-91).

3-There appears to be a YPet image overlaid in FOV#3 in Supp Fig. 14 It should be removed.

We have fixed the issue in the revised figure (Supplementary Fig. 14).

4-While I appreciate the authors' efforts to demonstrate some biological applications of their split-FPs in response to reviewer 2 and to my own comments, I find the data on the nuclear localization of zyxin to be of limited interest, and not representative of the advantages that split-FP probes might provide. For instance, it is not clear to me why such an observation could not have been made with traditional full FP fusions. Have the author also studied the nuclear localization of zyxin with full FP fusions? If so, why such an accumulation would be "revealed" by split-FP and not full FPs? The authors need to be much clearer on these points and they should provide some explanation. From a quick survey of the literature, it appears that the accumulation of zyxin in the nucleus has been observed by many other groups (see ref. 23 cited by the authors and other references therein). Specifically, in the case of ref. 23, such accumulation depends on the inhibition of CRM1-mediated nuclear exportation by leptomycin B. Why would tagging zyxin with split-FPs induce a similar nuclear accumulation? Does split-FP tagging prevent a normal function of nuclear export signals within zyxin, thus phenocopying the effect of leptomycin? In the absence of further explanation and discussion, these results have, unfortunately, very little interest and fail at conveying the advantages of using split-FP fusions

for biological imaging. In fact, as currently described, the zyxin data raise more questions on the use of split-FPs for protein tagging than they provide answers.

Based on this reviewer's suggestion, we have added the number of cells exhibiting the nuclear localization of full-length FP-tagged zyxin (page 10, lines 198-200). Using the same measurement in quantifying FP₁₁-tagged zyxin, we found that 50 out of 174 U2OS cells displayed the nuclear localization of full-length FP fusions, suggesting no apparent difference between the two labeling approaches (i.e., split FP vs. full-length FP). In the current demonstration of multiplexed labeling with split FPs (Fig. 4g), we over-expressed genetic fusions in single cells. We believe it is possible to extend this approach to label these endogenous proteins, which prevents over-expression artifacts (e.g., aberrant organelle and/or cellular morphology), via CRISPR/Cas9-mediated gene knock-in (Leonetti et al. *Proc Natl Acad Sci USA*, 113(25), 3501-8, (2016)). Because split FP tags are ~10 times smaller than the size of a full-length FPs, we can commercially synthesize short donor oligos, and thus omit multiple cloning steps of generating donor constructs. On top of that, a short donor can be inserted to the host genome with ~ 2.5 times higher recombination efficiencies than a full-length FP donor (Paix et al. *PNAS*, 114(50), 10745-10754, (2017)). Such a simple and efficient approach could be advantageous over full-length FP knock-in (Kamiyama et al., *Nat Commun*, 7, 11046 (2016)), ultimately facilitating the generation of multiple insertions in single cells. We have included this discussion into the main text (pages 10-11, lines 198-213). As Reviewer #2 recommended, we have combined Results and Discussion and we hope that this arrangement makes this part of manuscript clearer.

REVIEWERS' COMMENTS:

Reviewer #3 (Remarks to the Author):

The manuscript has been further improved by the implementation of requested changes and I only have 2 final comments:

1- Although not critical at this point, I don't think the author fully address the question of demonstrating key biological applications specific to the split-FP they have developed. The fact that a nuclear localization of zyxin is observed even with full-length FPs somewhat undermines the use of split-FPs altogether. While I understand that, in the long term, endogenous labeling via CRISPR might potentially provide some advantages, those are not fully demonstrated here.

2- In response to question 2 of reviewer 2 related to briefly discussing "various microscopy applications" in introduction, the authors cite their own work and largely fall short of providing a brief overview of the innovative microscopy applications born from split-GFP technologies (e.g. GRASP and single molecule imaging in *C. elegans*, tripartite assemblies, analyses of contact sites between organelles, studies of host/pathogen interactions, etc...). The authors would be well-advised to read recent reviews on split-FP applications (e.g. Pedelacq and Cabantous, doi: 10.3390/ijms20143479; Romei and Boxer, doi: 10.1146/annurev-biophys-051013-022846) and get acquainted with work preceding their own contribution to the field. Beyond focusing on specific multicolor applications with split-FPs, it is in the interest of the authors to also provide introductory examples showing that the technology their work with is applicable to a large variety of research domains.

Please see our responses to the comments in blue text.

Reviewers' comments:

Reviewer #3 (Remarks to the Author):

The manuscript has been further improved by the implementation of requested changes and I only have 2 final comments:

1- Although not critical at this point, I don't think the author fully address the question of demonstrating key biological applications specific to the split-FP they have developed. The fact that a nuclear localization of zyxin is observed even with full-length FPs somewhat undermines the use of split-FPs altogether. While I understand that, in the long term, endogenous labeling via CRISPR might potentially provide some advantages, those are not fully demonstrated here.

We greatly appreciate your comments. We cannot perform additional experiments at this moment; however, we plan on endogenously labeling multiple proteins with split FPs in a future experiment.

2- In response to question 2 of reviewer 2 related to briefly discussing "various microscopy applications" in introduction, the authors cite their own work and largely fall short of providing a brief overview of the innovative microscopy applications born from split-GFP technologies (e.g. GRASP and single molecule imaging in *C. elegans*, tripartite assemblies, analyses of contact sites between organelles, studies of host/pathogen interactions, etc...). The authors would be well-advised to read recent reviews on split-FP applications (e.g. Pedelacq and Cabantous, doi: 10.3390/ijms20143479; Romei and Boxer, doi: 10.1146/annurev-biophys-051013-022846) and get acquainted with work preceding their own contribution to the field. Beyond focusing on specific multicolor applications with split-FPs, it is in the interest of the authors to also provide introductory examples showing that the technology their work with is applicable to a large variety of research domains.

Thanks for the reviewer's feedback. We have included the two references (Romei et al., *Annu Reb Biophys* 48, 19-44 (2019); Pedelacq et al., *Int J Mol Sci* 20, (2019)) that this reviewer has recommended. In addition, we have modified the text of the Introduction:

(Page 3, lines 24-27) "*The GFP₁₁ fragment has been used in numerous biological studies^{3,4}: targeting nanomaterials in cells^{5,6}, forming protein oligomeric structures^{2,7}, verifying aggregation processes⁸, and imaging protein localization in living cells⁹.*"